# Effect of Partial Substitution of Sodium Chloride (NaCl) with Potassium Chloride (KCl) Coupled with High-Pressure Processing (HPP) on Physicochemical Properties and Volatile Compounds of Beef Sausage under Cold Storage at 4 °C

Theodora Ojangba [1,2], Li Zhang [1,*], Solomon Boamah [1], Yanglei Gao [1], Zhuo Wang [1]
and Francis Kweku Amagloh [2]

1 College of Food Science and Engineering, Gansu Agricultural University, Lanzhou 730070, China;
theodora@st.gsau.edu.cn (T.O.); solomon@st.gsau.edu.cn (S.B.); yanglei@st.gsau.edu.cn (Y.G.);
zhou@st.gsau.edu.cn (Z.W.)
2 Department of Food Science and Technology, University for Development Studies, Tamale 34983, Ghana;
fkamagloh@uds.edu.gh
* Correspondence: zhanglwubd@163.com; Tel.: +86-136-6935-2751

**Abstract:** This study aimed to evaluate the effects of partial substitution of sodium chloride (NaCl) with potassium chloride (KCl) in combination with high-pressure processing (HPP) on the physicochemical properties and volatile compounds of beef sausage during cold storage at 4 °C. Significant differences were found in the volatile compounds of beef sausages with 0%, 25%, and 50% NaCl contents partially substituted with KCl subjected to 28 days of storage and were well-visualized by heat map analysis. A total of 75 volatile compounds were identified and quantified in the beef sausages at the end of 28 days of storage, including 12 aldehydes, 4 phenols, 2 ketones, 18 alcohols, 8 acids, 3 esters, 14 terpenes, and 14 alkanes. Thirteen compounds had low odor activity values (OAV) (OAV < 1); however, high OAV (OAV > 1) were obtained after partial substitution of NaCl by KCl at 25% and 50% with HPP treatment compared to the non-HPP treated samples. In addition, 50% NaCl substitution with KCl in conjunction with HPP treatments increased thiobarbituric acid reactive substances (TBARS) by (0.46 ± 0.03 mg/MDA) compared with no HPP treatments. Replacement of 25% and 50% NaCl with KCl decreased TBARS by an average of 10.8% and 11.10%, respectively, compared to 100% NaCl coupled with HPP beef sausages. In summary, HPP and partial substitution of NaCl with KCl at 25% and 50% can be used to compensate for the reduction of NaCl in beef sausage by keeping the physical and flavor fraction at required levels.

**Keywords:** beef sausage; salt-replacement; high-pressure processing; protein digestion; protein and lipid oxidation; physicochemical characteristics; volatile compounds

## 1. Introduction

Currently, the available alternatives to reduce the salt (NaCl) content in meat products are limited and hence more methods need to be developed. Based on the structuration effects of high pressure (HP), the use of HPP could contribute to NaCl reduction in meat products [1–4]. Although the objective of HPP is mostly used for shelf-life extension and cold-pasteurization, pressurization can also be used for structuring and improving the functionality of meat products. It has been hypothesized that HPP could have analogous effects on the solubilization of myofibrillar proteins as salt and phosphates as a result of the reformation of protein spatial structure [5,6]. HPP enhanced the solubilization of myofibril at 150 MPa [7], and the reverse effects were observed with pressure above 300 MPa [6]. A severe denaturation of the proteins lowers the solubilization of proteins and the WHC. Additionally, previous studies have pointed out that HPP treatment improves the saltiness perception in meat products [8,9]. Tamm et al. [10] examined and developed

an industrial process to produce a salt-reduced cooked ham coupling HPP at different stages of the process with the salt replacer KCl. HPP treatments at different levels (100, 300, and 600 MPa at room temperature for 5 min) were applied at different processing steps (raw material, after injection, after tumbling, and after cooking). The cooking loss increased significantly with the pressure level (300–600 MPa); particularly, a pressurization higher than 300 MPa of the raw material and after the brine injection showed strong negative effects on the cooking loss. Bertram et al. [11] described the application of a very low-pressure treatment (7 MPa at 4 s) and the effect on muscle structure, water binding, and sensory properties in cured ham. They observed that the application of pressure showed strong evidence of alteration of these properties with the potential to reduce the critical amount of salt required in the products. In cooked meat and at the lowest NaCl concentration, the pressured meat attained a higher score in salt taste and juiciness than the tumbled meat. HPP is a beneficial alternative to thermal pasteurization after product manufacture in the meat processing industry [12] to prevent post-processing contamination. HPP has been tested for its effects on microbiological safety, and the quality of both low-salt beef sausages and cooked ham [13,14] and some quality indicators, such as lipid oxidation, color, and sensory characteristics, showed no significant changes. However, the effects of partial replacement of NaCl with KCl in combination with HPP on the quality characteristics of beef sausages during cold storage are less well-documented.

According to some studies, fermented meat products account for about 20–30% of salt consumption in a person's diet [15,16]. Excessive NaCl consumption, in turn, has been associated with an increased risk of hypertension, stroke, and vascular disease [17]. Beef sausages are a common cooked meat product worldwide, making them an ideal target to reduce salt consumption in the population. The most widely used methods to reduce NaCl content in fermented meat products are direct NaCl reduction and the use of salt substitutes to replace NaCl [18]. Several studies [14,15] have found the replacement of NaCl in foods such as meat products, which are of benefit to humans and the economy [19].

Food odor is directly related to volatile molecules emitted from meals, and they can be used to determine quality and safety. Simultaneous distillation and extraction (SDE) [20] and solid-phase microextraction (SPME) [21] have all been used. There is no ideal method for the isolation of volatile compounds from food. Moreover, each method is flawed in the extraction of compounds [22], resulting in different volatile profiles of the same product [23]. Given the growing interest in beef sausage products, there is a need to investigate whether NaCl content can be reduced without the addition of salt substitutes [24]. Some works look at the impact of direct salt reduction in different meat products such as bacon, ham, and salami, concluding that salt content can potentially be reduced without significantly affecting overall acceptability [11,25]. However, there are fewer findings on the impact of partially replacing 50% NaCl in beef sausages with KCl in combination with HPP on quality parameters during cold storage. Extreme HPP treatment increases protein denaturation and lipid oxidation, which negatively affects meat products qualities such as volatile compositions during storage. To acquire a better understanding of the impacts of the aforementioned treatments, further study is required to evaluate the effects of partial substitution of 50% NaCl by KCl in combination with HPP on the physicochemical properties and volatile compositions in beef sausages. Therefore, this study aims to avoid extreme protein denaturation and lipid oxidation to improve the physicochemical characteristics and volatile composition of beef sausage stored at 4 °C for 28 days by partial replacement of 50% NaCl with KCl combined with HPP (100 MPa for 5 min at 25 °C).

## 2. Materials and Methods

### 2.1. Experimental Design for Reduced-NaCl Beef Sausage

The extreme vertices mixture design was used to create the two-component salt combination (NaCl and KCl) in each beef sausage formulation according to McLean and Anderson [26]. In summary, the salt mixtures were generated by Design-Expert, a statis-

tical package (Statease Inc., Minneapolis, MN, USA). Each salt mixture was employed to substitute NaCl (a sum of 100 percent in an experimental mixture design) of the actual beef sausage as described in Table 1. The study was designed in two stages: in the first stage, different levels of salt replacement were used to select the best salt level based on several preliminary sensory tests (data not provided). Minced beef samples were randomly divided into three groups before the salting stage. Treatments were control (100% of NaCl), Treatment 1 (75% NaCl and 25% KCl), and Treatment 2 (50% NaCl and 50% KCl). In the second stage, each treatment group was further divided into two; each part was subjected to or without high-pressure processing at 100 MPa for 5 min at 25 °C. In all, six (6) treatment groups of sausages were used, with each group containing 35 chubs of beef samples (length 10 cm and weight 20 g). The experimental ranges for these two components were established following the US Food and Drug Administration's "reduced sodium" content claims (CFR—Code of Federal Regulations Title 21, 2021), which stated that this claim may be used on food labels if the food contains 25% less sodium than the initial or customary formulation.

**Table 1.** Percentages of sodium chloride and potassium chloride used in the formulation of beef sausage.

| First Stage | | | |
|---|---|---|---|
| **Ingredients** | **Treatments (%)** | | |
| | **Control** | **T1** | **T2** |
| Beef lean meat | 75 | 75 | 75 |
| Beef back fat | 12.39 | 12.39 | 12.39 |
| Water (cold ice) | 10 | 10 | 10 |
| Spices | 0.4 | 0.4 | 0.4 |
| Phosphate | 0.2 | 0.2 | 0.2 |
| $NaNO_2$ | 0.01 | 0.01 | 0.01 |
| Sodium chloride (NaCl) | 2 | 1.5 | 1 |
| Potassium chloride (KCl) | - | 0.5 | 1 |
| Total | 100 | 100 | 100 |
| Second Stage | | | |
| HPP | 100% NaCl | 75% NaCl and 25% KCl | 50% NaCl and 50% KCl |
| No HPP | 100% NaCl | 75% NaCl and 25% KCl | 50% NaCl and 50% KCl |

Control (100% NaCl-No HPP); T1 (75% NaCl and 25% KCl-No HPP); T2 (50% NaCl and 50% KCl-No HPP) and control (100% NaCl-HPP); T1 (75% NaCl and 25% KCl-HPP); T2 (50% NaCl and 50% KCl-HPP).

## 2.2. Reduced-NaCl Beef Sausage Preparation

Six kilograms (6 kg) of fresh beef (semimembranosus muscles) and 1 kg beef back fat was purchased from a local BHG supermarket (Lanzhou, China). Fresh beef and fat were vacuum-packed and stored at −18 °C until required for sausage production. The frozen meat and fat was allowed to thaw slightly before being minced. The meat was trimmed of excess fat and minced through a 5 mm plate using a vacuum mincer (BX150, Hengshun Machinery Factory, Jinan, Shandong, China) at 75 rpm for 3 min. Application of HPP was performed according to Yang et al. [27] with modifications. Water was used as the pressure-transfer medium, and the entire system was cooled to an initial ambient temperature of 25 °C by a thermos-stated jacket. The vacuum-packed (Djpack, DZ-450A, Wenzhou Dajiang Vacuum Packaging Machinery Co., Ltd., Wenzhou, China) meat batters mixed with the salts only contained in vacuum pouches of about 150 g in weight were subjected to HP treatment at 100 MPa for 5 min in an isostatic press (HPP. L2-600/1, Tianjin Huatai-Senmiao Bioengineering Technology Co., Ltd., Tianjin, China). After HP treatment, the meat samples were immediately removed and placed in a bowl chopper, and ingredients such as phosphate salt, garlic, nutmeg, black and white pepper, and chili were added. Six (6) treatment groups of sausages were prepared including the control (100% NaCl-No HPP), T1 (75% NaCl and 25% KCl-No HPP); T2 (50% NaCl and 50% KCl-No HPP) and control

(100% NaCl-HPP), T1 (75% NaCl and 25% KCl-HPP), and T2 (50% NaCl and 50% KCl-HPP), with each group containing 35 chubs of beef samples (length 10 cm and weight 20 g).

The meat mixture was then kept refrigerated overnight. The next day, each treatment batch was stuffed into pre-soaked collagen casings (19 mm diameter, UniPac, Edmonton, AB) (Handtmann, model VF80, Waterloo, ON, Canada), and the casings were stretched and sealed. Five chubs from each recipe were cut into 3 mm thick slices, which were vacuum-packed for further analysis (five slices per pack), in high-barrier Mylar/polyethylene bags (Ulma TF-Supra Packaging Machine Co., Ltd., Tianjin, China). The remaining chubs in each formulation were vacuum-packed as a whole.

All vacuum-packed samples were randomly divided into two groups for high-pressure treatment (HPP): control (no HPP) and HPP. The HPP batches were immediately subjected to a pressure of 100 MPa for 5 min at 25 °C using 850 Mini FoodLab high-pressure vessels with a capacity of 0.3 l (Stansted Fluid Power Ltd., Harlow CM19 5FN, UK). Subsequently, all samples were randomly assigned to subgroups for storage intervals (0, 3, 7, 14, 21, and 28 days), packed into boxes, and stored at 4 °C until analysis (Figure 1).

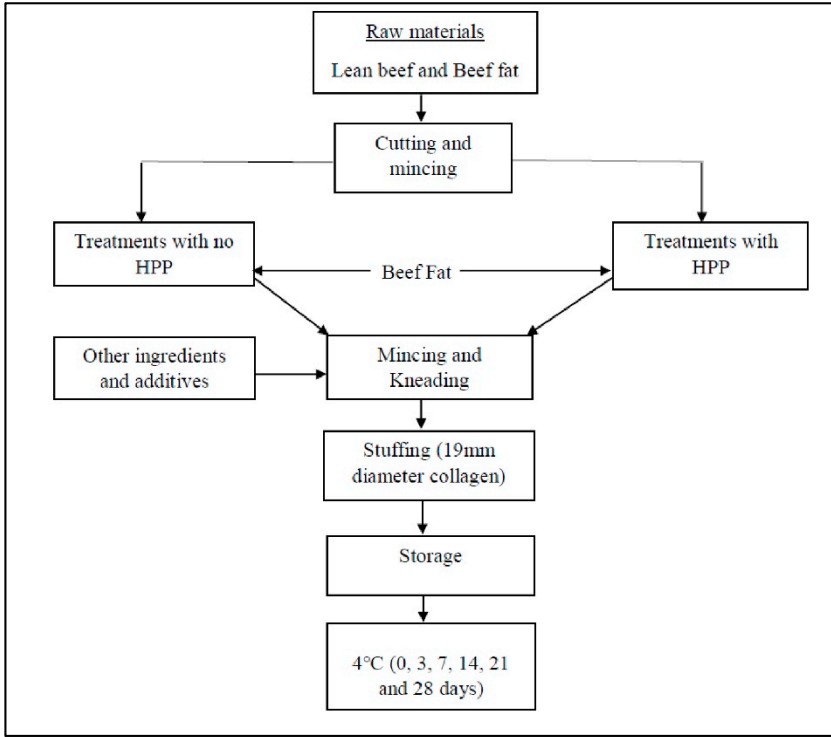

**Figure 1.** Flow diagram of the manufacture and storage of beef sausage under study (treatments with HPP and treatment with no HPP).

## 2.3. Determination of Water Activity (aw)

For water activity (aw) determination, samples were randomly collected and analyzed from each treatment on 0, 3, 7, 14, 21, and 28 storage days (time) at 4 °C. Aw of raw beef sausages was determined by direct reading on an HD-3A intelligent water activity meter (Decagon Devices, Inc., Pullman, WA 99163, USA) for 20 min. All analyses were performed in triplicate according to [28] with slight modifications. This was repeated three times with three replicates.

## 2.4. pH Determination

The pH values of all raw sausages were measured after 0, 3, 7, 14, 21, and 28 days of storage in a homogenate prepared from 5 g of the sample and distilled water (20 mL) using an acidimeter (PHS-320) with a combined electrode (glass body with spear tip) with

temperature compensation according to Sun et al. [29]. All determinations were performed in triplicate. This was repeated three times with three replicates.

### 2.5. Color Measurement

The color was measured using a Minolta handheld spectrophotometer (Minolta CR300b, Suita-Shi, Osaka, Japan) with a 10 °C observer angle and illuminant D65, calibrated against a white tile immediately before readings were taken. Color measurements (CIE *L*\*, *a*\*, and *b*\*) were made for each after 0, 3, 7, 14, 21, and 28 days of storage in a retail display case (4.0 ± 1.0 °C) under 24 h fluorescent lighting with an average intensity of 1630 lx. All measurements were taken in duplicate, with a 90° clockwise sample rotation between measurements. This was repeated three times with three replicates.

### 2.6. Water-Holding Capacity Analysis

Water-holding capacity (WHC) was measured according to the procedure described by Monetti [30] with slight modifications, and values were expressed as water excreted from the meat. After 0, 3, 7, 14, 21, and 28 days of storage, $2 \times 1$ cm$^2$ of sausage with the outer casing removed was measured by recording the initial weight (*W*1) of three samples for each batch and placing them between two cheesecloths sandwiched between two Whatman paper filters (55 mm diameter; Whatman, Clifton, NJ, USA). Samples were pressed for 5 min with a 1 kg metal cylinder, and the final weight was also determined (*W*2). The difference between the initial weight (before) and the final weight (after) pressing was calculated as a percentage using the following formula:

$$\frac{W1 - W2}{W1} \times 100\%$$

### 2.7. Lipid Oxidation (TBARS) Analysis

Lipid oxidation was measured using the 2-thiobarbituric acid reactive substances (TBARS) assay [31] modified for interference with nitrites in cured meat [32]. Briefly, 2.5 g of sausage sample was mixed with 10 mL of reverse osmosis water (RO) and 1.0 mL of sulfanilamide reagent (0.5% sulfanilamide in 20% HCl (*v*/*v*)) in a Magic Bullet blender and ground for 30 s. The mixture was then mixed with sulfanilamide reagent. The mixture was placed in a round-bottomed boiling flask along with 10.0 mL of RO water used to wash the mixer. A total of 2.0 mL of 4 N HCl was added to bring the mixture to a pH of 1.5. Glass beads (6) were added to the boiling flask, which was then brought to a vigorous boil on a hot plate set at 375 °C (Corning PC-620D, Corning Incorporated Life Sciences, Tewksbury, MA, USA). A simple glass distillation apparatus (hot plate, boiling flask, short vertical column connected to an inclined horizontal water-cooled condenser) was used to collect ~10 mL of the distillate. A total of 5.0 mL of the distillate was mixed with 5.0 mL of TBA reagent (0.02 M 2-thiobarbituric acid in 90% glacial acetic acid) in a 10 mL screw-capped glass tube and vortexed for 10 s at maximum speed (VWR Vortexer 2, VWR International Co., Edmonton, AB, Canada). The tubes were then immersed in a boiling water bath for 35 min and then cooled in tap water for 10 min. The absorbance at 532 nm of the resulting solution was measured using a spectrophotometer (EPOCH2 Plate Reader, BioTek, Chicago, IL, USA. TBARS values were determined against a standard linear curve of malonaldehyde standard solution (1,1,3,3-tetra-ethoxypropane). TBARS values were expressed as milligrams malonaldehyde equivalents/kilogram meat sample. Measurements were performed in triplicate for each treatment. This was repeated three times with three replicates.

### 2.8. Protein Carbonyl Analysis

The concentration of protein carbonyls was determined by derivatization with DNPH (2,4-dinitrophenylhydrazine) as described by Levine et al. [33] with some modifications. An aliquot of approximately 2 mg of lyophilized MPI was dissolved in 500 μL of 6.0 M guanidine hydrochloride in 20 mM potassium dihydrogen phosphate (pH 2.3) for 1 h in

a water bath heated to 50 °C. Dissolved samples (S) were incubated with 500 μL 10 mM DNPH dissolved in 2.0 M HCl, and dissolved blank samples (B) were incubated with 500 μL 2.0 M HCl instead of the DNPH solution. Samples and blanks were heated at 37 °C for 1 h in water bath and were shaken every 10 min. Then, 325 μL of 50% TCA was added to the samples and blanks, shaken for 30 s, and placed on ice for 10 min before centrifugation at 2300× *g* for 10 min, after which the supernatant was discarded. Excess DNPH was removed by washing with 1.0 mL ethanol ethyl acetate (1:1) containing 10 mM HCl, vortexing, allowing to react for 10 min, and then centrifuging at 2300× *g* for 10 min. This was repeated three times and, after each wash, the supernatant was discarded. After the last wash, excess solvent was removed by gently rinsing the pellets with air. A total of 1.0 mL of 6.0 M guanidine hydrochloride in 20 mM potassium dihydrogen phosphate (pH 2.3) was used to dissolve the pellets. The samples were then placed in a water bath heated to 37 °C for 30 min. The final solution was centrifuged at 2300× *g* for 10 min to remove insoluble material. The carbonyl concentration (nmoL/mg protein) was calculated from the absorbance at 280 and 370 nm of the samples as described below using an absorbance coefficient at 370 nm of 22,000 $M^{-1}$ $cm^{-1}$ for the hydrazones formed [34].

$$\frac{C_{hydrazone}}{C_{protein}} = \frac{A_{370} \times 10^6}{\text{hydrazine}_{370}(A_{280} - A_{370} \times 0.43)}$$

The contribution obtained from the blanks (B) was subtracted from the contribution obtained from the corresponding samples.

*2.9. In Vitro Digestibility*

Myofibrillar protein digestibility was assessed according to the formulation of Sante-Lhoutellier et al. [35] with slight modifications. Two grams (2 g) of beef sausage from each treatment group was homogenized using 20 mL of phosphate buffer (pH 6.8).

Gastric pepsin (1000 U/mg), pancreatic trypsin (from porcine pancreas 250 U/mg), and α-chymotrypsin (1000 U/mg) were obtained from Solarbio Science & Technology Co., Ltd., Beijing, China.

The proteins were suspended in 33 mM glycine buffer with a pH of 1.8 (gastric pH), and the final concentration was adjusted to 0.8 mg/mL in the same buffer. The proteins were first digested with gastric pepsin (enzyme: protein = 1:100, *w/w*) for 90 min at 37 °C. Digestion was terminated by the addition of 15% (final concentration) trichloroacetic acid (TCA) with a reaction time of 10 min. After centrifugation at 4000× *g* for 10 min, the content of hydrolyzed peptides in the soluble fraction was determined using a spectrophotometer at 280 nm, and the proteolysis rate was expressed in optical density units per hour (ΔOD/h). The insoluble fractions of the 90 min pepsin hydrolysate were washed twice in 33 mM glycine buffer at pH 8.0 (duodenal pH), and the final concentration was adjusted to 0.8 mg/mL. Proteins were then hydrolyzed for 60 min at 37 °C by a mixture of trypsin and α-chymotrypsin (enzyme:protein = 1:100 and 1:1000, *w/w*). The digestion was terminated by adding 15% (final concentration) TCA with different reaction times at 10 min, and the rate of proteolysis was measured at 280 nm using a spectrophotometer (EPOCH2 Plate Reader, BioTek, Chicago, IL, USA). This was repeated three times with three replicates.

*2.10. Volatile Compound Analysis*

Volatile compounds in the beef sausages were extracted by solid-phase micro extraction (SPME) and analyzed by gas chromatography/mass spectrometry (GC/MS) as described by Wen et al. [36] with slight modifications. To evaluate the contribution of these compounds to the flavor profile of the sausages, the odor activity value (OAV) was calculated by dividing the volatile compound content by their threshold values from the database (https://www.vcf-online.nl/VcfCompoundSearch.cfm accessed on 25 November 2021). This was repeated three times with three replicates.

*2.11. Gas Chromatography–Mass Spectrometry*

Volatile compounds were identified and quantified by the headspace/solid-phase microextraction (HS-SPME) and gas chromatography coupled to mass spectrometry (GC-MS) according to [37] with slight modifications. A triple SPME fiber of DVB/CAR/PDMS 50/30 μm (conditioned at 270 °C/30 min), was exposed for 1 h and 30 min at 40 °C. In a 20 mL capped vial, 10 g of minced beef sausages, 2 g of NaCl, and 5 μL of internal standard (8.82 ppm, 2-octanol) were added. The vial was tightly capped and equilibrated in a water bath at 90 °C for 30 min. The volatile compounds were transferred to the GC injector and desorbed (260 °C/5 min). The analysis was performed in a spitless mode (50 mL/min during 2 min) on a 6890 N Agilent gas chromatography coupled to a 5973 N Agilent Mass Detector. The selected chromatography conditions were a polar column (DB-WAX, 60 m × 0.25 mm id; 0.25 μm film thickness), helium as the carrier gas (1 mL/min), and oven temperature started at 40 °C/3 min, ramped at 3 °C/min to 150 °C, and up to 220 °C (7 °C/min for 5 min). The MS operated with electron energy of 70 eV using the electron impact mode, the ion source temperature was 230 °C, and the scanning was carried out from 45 to 550 amu. Compounds were identified by comparison with mass spectra and the linear retention index (RI) with spectral data from NBS75K and Wiley G 1035 libraries and calculating linear retention indices (LRI) relative to a range of alkanes ($C_1$–$C_{19}$). The sums of the abundances of up to four characteristic ions per compound were used for semi-quantitative determination. Abundances of volatile compounds were referenced to the internal standard (IS) (compound peak area multiplied by $10^3$ and divided by IS peak area). This was repeated three times with three replicates. Finally, concentrations of volatile compounds were obtained and expressed as ng/g.

*2.12. Statistical Analysis*

All specific experiments were carried out in triplicates, and the data were expressed as the mean ± SD values. The results were subjected to analysis of variance (ANOVA) using SPSS Win 12.0 software (SPSS Inc., Chicago, IL, USA). Differences between treatments were analyzed using the least significant difference test, and significance was defined at $p < 0.05$. Significant differences ($p < 0.05$) among means were identified using Duncan's multiple range tests using two-way ANOVA.

## 3. Results

*3.1. pH and Water-Holding Capacity*

Partial replacement of NaCl with KCl in conjunction with HPP had a significant ($p < 0.05$) effect on both pH and WHC of beef sausage during cold storage. Across the six (6) days of storage (0, 3, 7, 14, 21, and 28), 100% NaCl combined with HPP increased pH and WHC by an average of (5.4 ± 0.03) and 54.70%, respectively, compared to the control (100% NaCl unpressurized) (Figure 2A,B). Similarly, replacement of 25% NaCl with KCl in combination with HPP increased pH and WHC by an average of (5.5 ± 0.00) and 58.13%, respectively, compared to 25% NaCl with KCl without HPP. However, 50% NaCl replacement by KCl combined with HPP decreased the pH and WHC by an average of 2.05% and 18.64%, respectively, compared with 50% NaCl replacement by KCl without HPP (Figure 2A,B).

*3.2. Color*

Partial substitution of NaCl in combination with HPP significantly affected sausage color ($p < 0.05$). Compared to samples treated with HPP from days 28 to 0, 50% NaCl replacement with KCl at day 0 decreased the degree of redness (*a\**) by (5.46 ± 0.23). However, 50% NaCl replacement with KCl in conjunction with HPP at day 0 increased yellowness (*b\**) by (26.66 ± 1.08) compared to 50% NaCl replacement with KCl in conjunction with HPP at day 28, which was (24.16 ± 1.07) (Table 2).

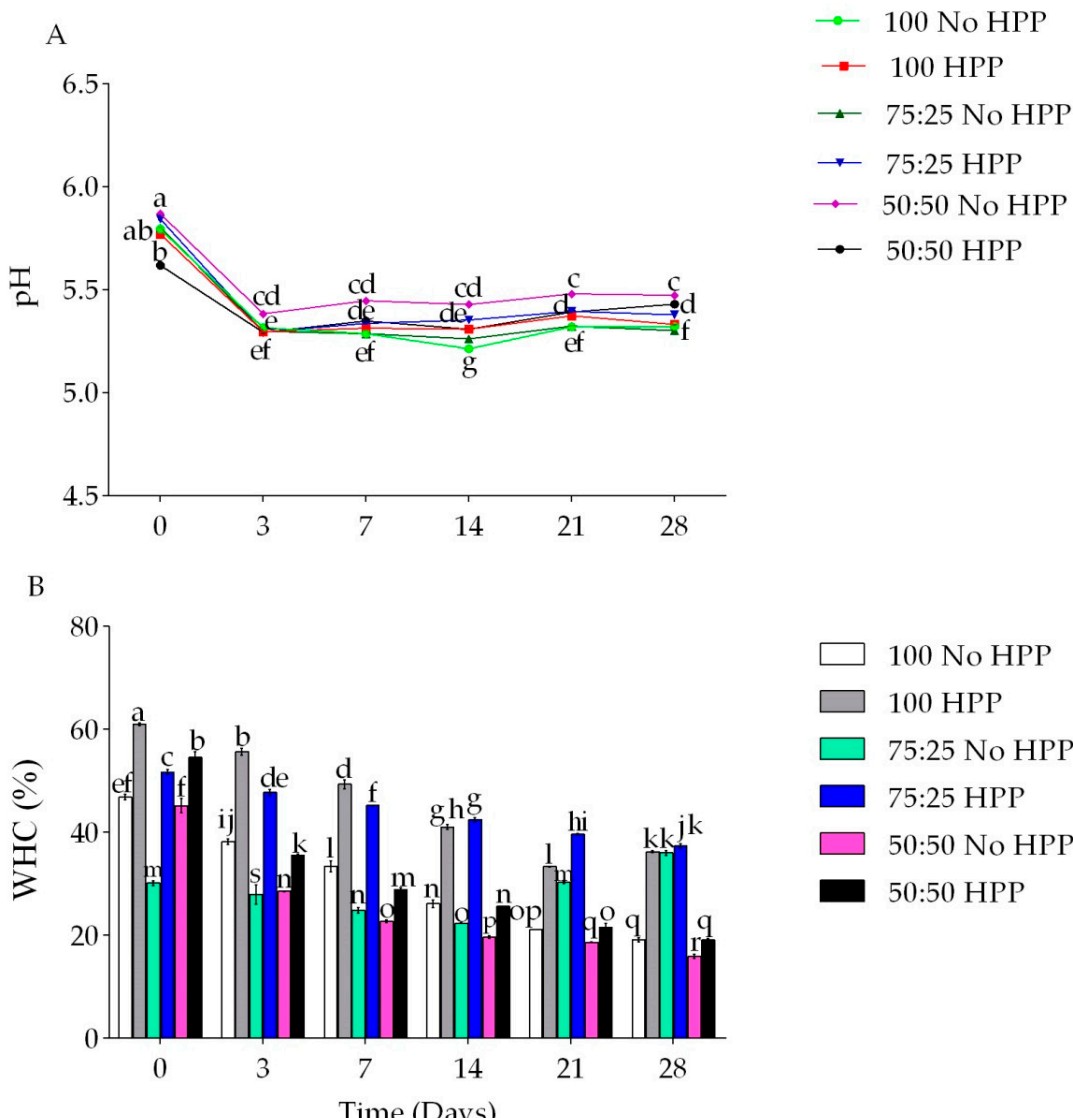

**Figure 2.** Effect of partial substitution of NaCl coupled with HPP on (**A**) pH and (**B**) water-holding capacity of beef sausage under different cold storage days. Data were presented in means ± SD. Different lower case letters indicate significant differences at *p* < 0.05 in Duncan's multiple range tests using two-way ANOVA.

### 3.3. Water Activity and Lipid Oxidation (TBARS)

Partial substitution of NaCl with KCl in combination with HPP significantly (*p* < 0.05) affected water activity (aw) and lipid oxidation (MDA). At 100% NaCl with no KCl replacement, the HPP treatment increased the aw by (0.08 ± 0.01 and 0.69 ± 0.01) on days 3 and 21, respectively, compared to the control (100% NaCl unpressurized). However, on day 3, the samples treated with HPP at 75% NaCl and 25% KCl increased the aw value by (0.54 ± 0.02) compared to the 75% NaCl and 25% KCl samples without HPP. An increase in KCl content increased the aw content. The 50% NaCl replacement with KCl coupled with HPP increased the aw content by (0.7 ± 0.01) at day 0 (Figure 3A).

The MDA content of samples treated with 100% NaCl and coupled with HPP increased the TBARS by an average of (0.54 ± 0.02 mg/MDA) compared to the control across the six storage days (Figure 3B). A comparison of 25% NaCl substitution with KCl coupled with HPP treatments to no HPP treatment increased MDA content across the six storage days by an average of (0.49 ± 0.02 mg/MDA). In addition, 50% NaCl substitution with KCl in conjunction with HPP treatments increased MDA content by an average of

(0.46 ± 0.03 mg/MDA) compared with no HPP treatments across the six storage days. Replacement of 25% and 50% NaCl with KCl decreased MDA content by an average of 10.8% and 11.10% respectively, compared to 100% NaCl HPP.

**Table 2.** Effect of partial substitution of NaCl coupled with HPP at 100 MPa on lightness, redness, and yellowness of beef sausage under different cold storage days.

| Treatments | Time/Days | $L^*$ | | $a^*$ | | $b^*$ | |
|---|---|---|---|---|---|---|---|
| | 0 d | Mean | SD | Mean | SD | Mean | SD |
| No HPP | 100%:0%(NaCl:KCl) | 57.82 [abc] | 0.61 | 7.58 [n] | 0.04 | 25.32 [bc] | 0.37 |
| | 75%:25%(NaCl:KCl) | 57.34 [bc] | 1.67 | 7.48 [n] | 0.04 | 25.52 [ab] | 0.78 |
| | 50%:50%(NaCl:KCl) | 59.80 [abc] | 1.49 | 8.56 [m] | 0.15 | 26.60 [a] | 1.17 |
| HPP | 100%:0%(NaCl:KCl) | 58.48 [abc] | 2.73 | 6.32 [o] | 0.66 | 24.88 [bcd] | 1.50 |
| | 75%:25%(NaCl:KCl) | 57.70 [abc] | 2.09 | 5.76 [p] | 0.13 | 24.86 [bcd] | 1.65 |
| | 50%:50%(NaCl:KCl) | 60.14 [abc] | 1.72 | 5.46 [p] | 0.23 | 26.66 [a] | 1.08 |
| | 3 d | $L^*$ | | $a^*$ | | $b^*$ | |
| No HPP | 100%:0%(NaCl:KCl) | 57.94 [abc] | 2.28 | 11.54 [ij] | 0.32 | 24.12 [cd] | 1.49 |
| | 75%:25%(NaCl:KCl) | 57.26 [bc] | 2.34 | 12.5 [fg] | 0.20 | 24.22 [bcd] | 0.79 |
| | 50%:50%(NaCl:KCl) | 58.68 [abc] | 2.34 | 13.32 [de] | 0.18 | 24.92 [bcd] | 1.36 |
| HPP | 100%:0%(NaCl:KCl) | 58.08 [abc] | 2.14 | 10.62 [k] | 0.25 | 25.02 [bcd] | 1.08 |
| | 75%:25%(NaCl:KCl) | 58.44 [abc] | 1.87 | 10.28 [k] | 0.22 | 25.22 [bcd] | 0.47 |
| | 50%:50%(NaCl:KCl) | 58.64 [abc] | 1.33 | 9.42 [l] | 0.36 | 24.84 [bcd] | 0.53 |
| | 7 d | $L^*$ | | $a^*$ | | $b^*$ | |
| No HPP | 100%:0%(NaCl:KCl) | 57.42 [abc] | 2.27 | 12.6 [fg] | 0.33 | 24.02 [cd] | 1.36 |
| | 75%:25%(NaCl:KCl) | 57.96 [abc] | 2.41 | 13.24 [de] | 0.27 | 24.20 [bcd] | 0.48 |
| | 50%:50%(NaCl:KCl) | 58.06 [abc] | 2.37 | 14.30 [ab] | 0.32 | 24.78 [bcd] | 0.29 |
| HPP | 100%:0%(NaCl:KCl) | 57.84 [abc] | 1.72 | 11.54 [ij] | 0.46 | 24.34 [bcd] | 0.69 |
| | 75%:25%(NaCl:KCl) | 58.06 [abc] | 2.28 | 11.62 [ij] | 0.15 | 24.60 [bcd] | 0.29 |
| | 50%:50%(NaCl:KCl) | 59.68 [abc] | 1.17 | 10.20 [k] | 0.16 | 25.28 [bcd] | 0.41 |
| | 14 d | $L^*$ | | $a^*$ | | $b^*$ | |
| No HPP | 100%:0%(NaCl:KCl) | 58.46 [abc] | 1.39 | 14.48 [ab] | 0.19 | 24.54 [bcd] | 0.94 |
| | 75%:25%(NaCl:KCl) | 58.82 [abc] | 2.68 | 13.66 [cd] | 0.13 | 24.76 [bcd] | 1.05 |
| | 50%:50%(NaCl:KCl) | 60.44 [ab] | 2.14 | 14.82 [a] | 0.40 | 25.14 [bcd] | 0.24 |
| HPP | 100%:0%(NaCl:KCl) | 58.82 [abc] | 1.91 | 12.36 [gh] | 0.27 | 24.46 [bcd] | 0.38 |
| | 75%:25%(NaCl:KCl) | 57.90 [abc] | 1.57 | 12.32 [gh] | 0.52 | 25.18 [bcd] | 0.74 |
| | 50%:50%(NaCl:KCl) | 59.78 [abc] | 2.50 | 11.46 [ij] | 0.25 | 24.84 [bcd] | 0.30 |
| | 21 d | $L^*$ | | $a^*$ | | $b^*$ | |
| No HPP | 100%:0%(NaCl:KCl) | 57.06 [c] | 1.40 | 12.58 [fg] | 0.04 | 23.92 [d] | 0.28 |
| | 75%:25%(NaCl:KCl) | 59.36 [abc] | 3.58 | 13.00 [ef] | 0.85 | 25.06 [bcd] | 1.18 |
| | 50%:50%(NaCl:KCl) | 57.90 [abc] | 2.79 | 14.28 [ab] | 0.23 | 24.76 [bcd] | 0.64 |
| HPP | 100%:0%(NaCl:KCl) | 57.52 [abc] | 2.81 | 11.54 [ij] | 0.22 | 24.44 [bcd] | 1.13 |
| | 75%:25%(NaCl:KCl) | 58.90 [abc] | 1.21 | 10.52 [k] | 0.37 | 24.96 [bcd] | 0.48 |
| | 50%:50%(NaCl:KCl) | 59.20 [abc] | 2.65 | 11.16 [j] | 0.95 | 25.12 [bcd] | 1.07 |
| | 28 d | $L^*$ | | $a^*$ | | $b^*$ | |
| No HPP | 100%:0%(NaCl:KCl) | 57.92 [abc] | 2.09 | 13.18 [de] | 0.22 | 24.08 [cd] | 0.66 |
| | 75%:25%(NaCl:KCl) | 58.08 [abc] | 1.23 | 13.92 [bc] | 0.34 | 24.52 [bcd] | 0.44 |
| | 50%:50%(NaCl:KCl) | 58.58 [abc] | 1.63 | 14.24 [b] | 0.17 | 24.22 [bcd] | 0.56 |
| HPP | 100%:0%(NaCl:KCl) | 60.74 [a] | 2.81 | 11.64 [ij] | 0.18 | 25.02 [bcd] | 0.31 |
| | 75%:25%(NaCl:KCl) | 59.54 [abc] | 0.93 | 11.48 [ij] | 0.27 | 24.64 [bcd] | 0.56 |
| | 50%:50%(NaCl:KCl) | 58.52 [abc] | 2.81 | 11.90 [hi] | 1.37 | 24.16 [bcd] | 1.07 |

Data are means ± SD of replicates and those in a column followed by different letters are significantly different at $p < 0.05$, based on Duncan's new multiple range test using two-way ANOVA.

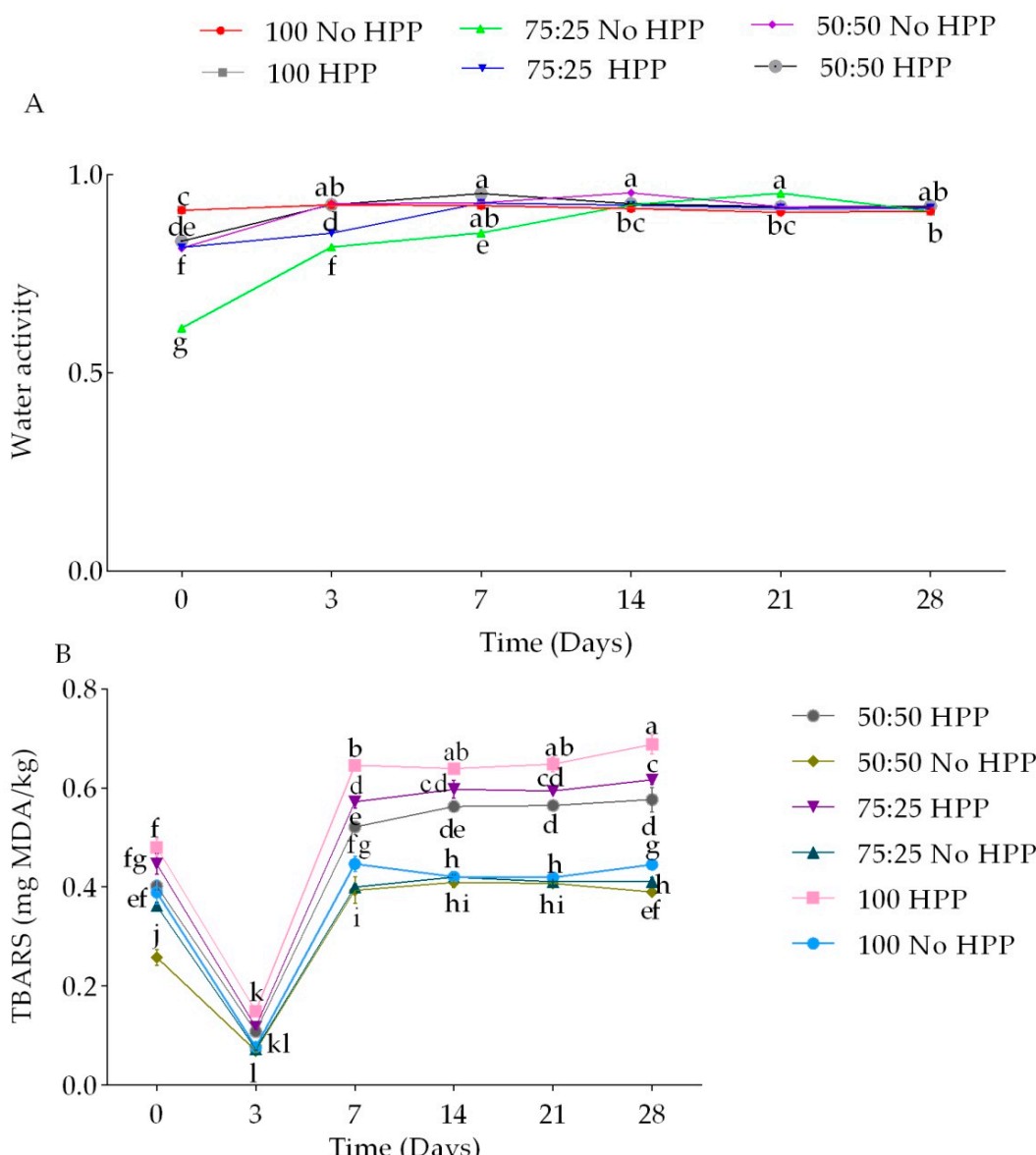

**Figure 3.** Effect of partial substitution of NaCl coupled with HPP on (**A**) water activity and (**B**) lipid oxidation of beef sausage under different cold storage days. Data were presented in means and SD. Different lower case letters indicate significant differences at *p* < 0.05 in Duncan's multiple range tests using two-way ANOVA.

*3.4. Carbonyl and Sulfhydryl Content*

At 100%, NaCl combined with HPP significantly (*p* < 0.05) increased the carbonyl content of beef sausages by an average of 2.78 ± 0.09 nmol/mg protein across the six (6) days of storage (0, 3, 7, 14, 21, and 28 days) compared to the control (100% NaCl unpressurized). Partial substitution of NaCl along with 100 MPa pressure treatment decreased protein carbonylation. On days 0 and 3, substitution of 25% NaCl with KCl in conjunction with HPP decreased the carbonyl content of beef sausages by 0.44 ± 0.03 nmol/mg protein and 0.59 ± 0.00 nmol/mg protein, respectively, compared to treatment without HPP. However, on days 7, 14, 21, and 28, the carbonyl content of beef sausages with 25% NaCl substitution by KCl in combination with HPP increased by 0.57 ± 0.02 nmol/mg protein, 0.60 ± 0.03 nmol/mg protein, 0.59 ± 0.00 nmol/mg protein, and 0.62 ± 0.02 nmol/mg protein compared to 25% NaCl substitution by KCl without HPP treatment (Figure 4A).

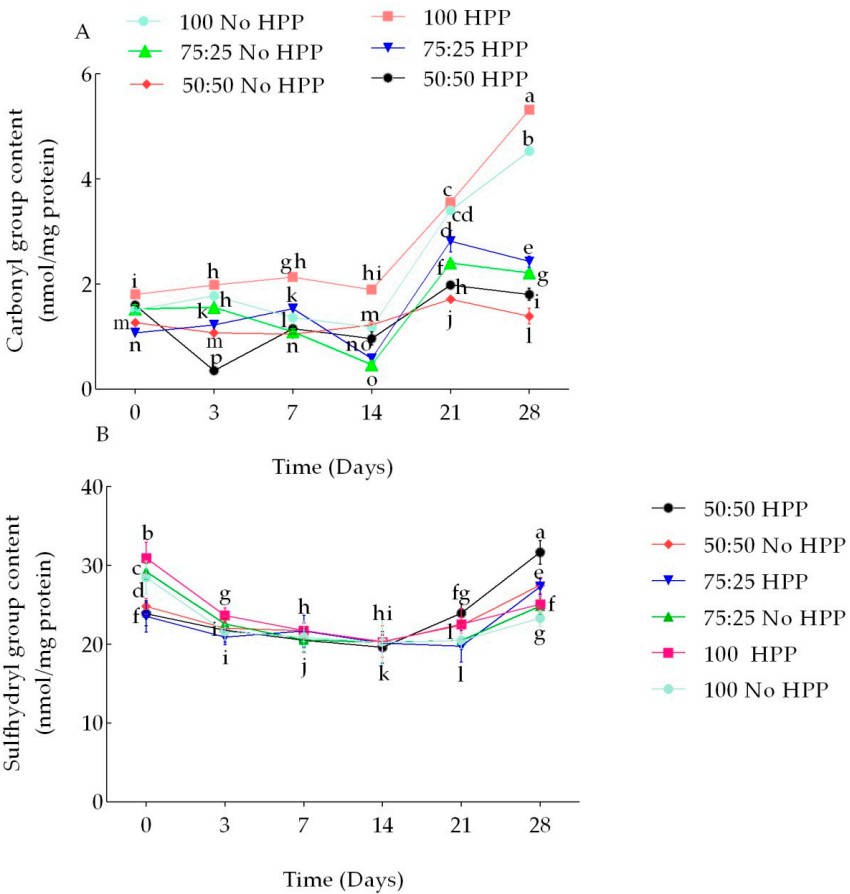

**Figure 4.** Effect of partial substitution of NaCl coupled with HPP on (**A**) carbonyl content and (**B**) sulfhydryl content of beef sausage under different cold storage days. Different lower case letters indicate significant differences at $p < 0.05$ in Duncan's multiple range tests using two-way ANOVA.

Similarly, in the present study, sulfhydryl content of beef sausages treated with HPP at 25% NaCl replacement with KCl decreased by $23.56 \pm 3.47$ nmol/mg protein, $20.92 \pm 1.74$ nmol/mg protein, $20.16 \pm 0.02$ nmol/mg protein, and $19.74 \pm 3.47$ nmol/mg protein on days 0, 3, 14, and 21, respectively, compared to samples without HPP treatment. On days 7 and 28, HPP increased sulfhydryl content by $21.68 \pm 1.75$ nmol/mg protein and $27.33 \pm 1.75$ nmol/mg protein, respectively, compared to the treatments with 25% NaCl and KCl without HPP (Figure 4B). However, in the beef sausages with 50% NaCl and KCl treated with HPP, sulfhydryl content decreased by $23.90 \pm 2.60$ nmol/mg protein, $21.77 \pm 2.43$ nmol/mg protein, $20.51 \pm 2.64$ nmol/mg protein and $19.60 \pm 3.48$ nmol/mg protein on days 0, 3, 7, and 14 compared to samples not treated with HPP. On days 21 and 28, beef sausages treated with HPP increased sulfhydryl content by $23.96 \pm 2.08$ nmol/mg protein and $31.63 \pm 2.08$ nmol/mg protein, respectively, compared to samples not treated with HPP.

### 3.5. In Vitro Digestion

In this study, replacement of 25% and 50% NaCl with KCl in combination with HPP increased pepsin digestion by an average of 26.73% and 45.67% compared to 100% NaCl with HPP during the six (6) days of storage (Figure 5A). With 50% NaCl replaced by KCl, beef sausages treated with HPP increased trypsin and α-chymotrypsin by 45.93% at day 3 compared to 100% NaCl without HPP at day 0 (Figure 5B). However, on days 21 and 28, 50% NaCl replaced by KCl combined with HPP beef sausages increased trypsin and α-chymotrypsin digestion by 93.4% and 87.53%, respectively, compared to samples not treated with HPP.

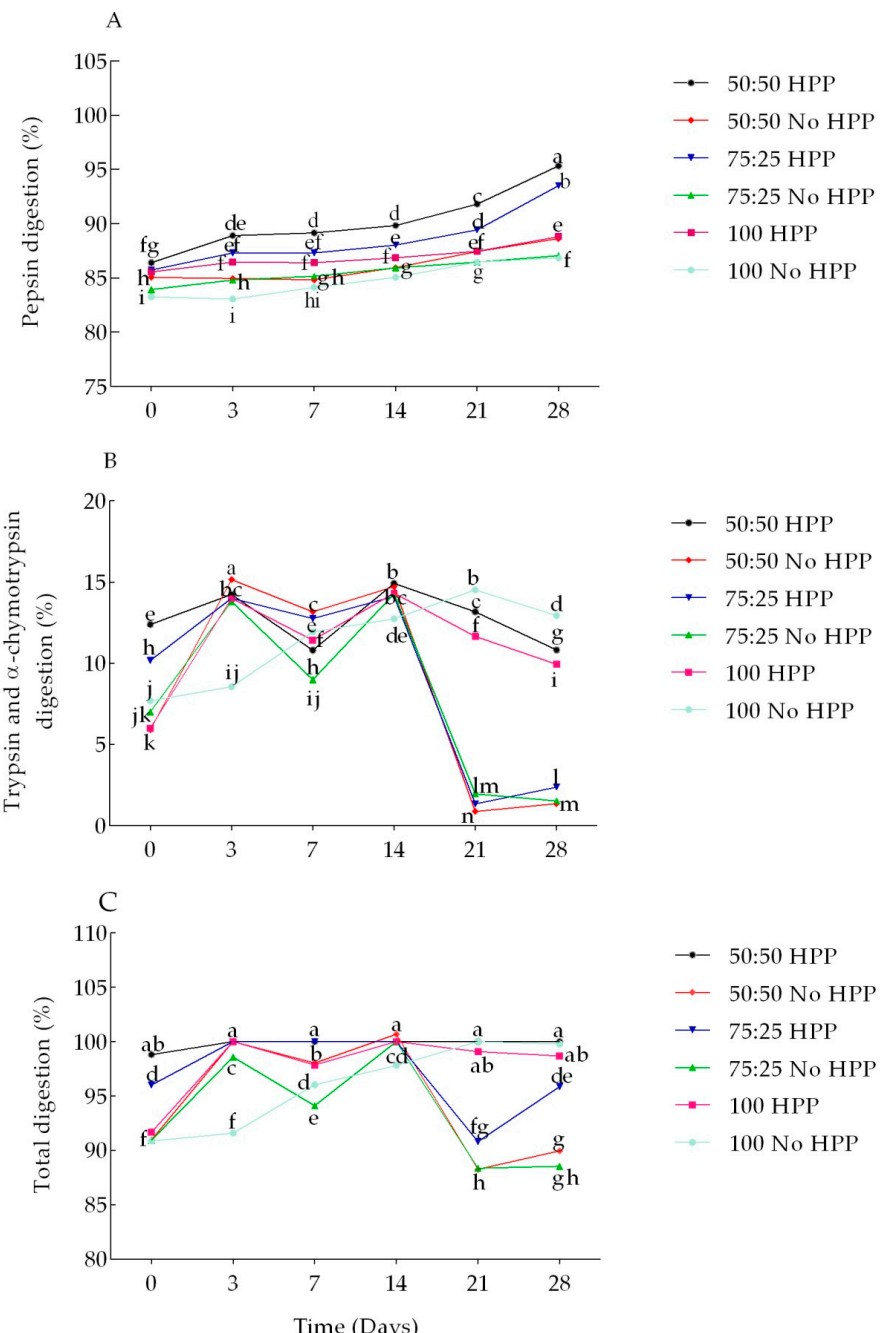

**Figure 5.** Effect of partial substitution of NaCl coupled with HPP on (**A**) pepsin digestion, (**B**) trypsin and *α*-chymotrypsin, and (**C**) total digestion of beef sausage under different cold storage days. Different lower case letters indicate significant differences at *p* < 0.05 in Duncan's multiple range tests using two-way ANOVA.

In addition, the samples with 25% NaCl substituted with KCl combined with HPP increased total digestibility by an average of 3.82% during the six (6) days of storage. Overall, the 50% substitution of NaCl with KCl in conjunction with HPP treatment increased total digestibility by an average of 51.77% compared to the 25% substitution of NaCl with KCl in conjunction with HPP treatment (Figure 5C).

### 3.6. Volatile Compounds

At the end of the 28-day storage period, 75 volatile chemicals were detected and measured in the beef sausages, including 12 aldehydes, 4 phenols, 2 ketones, 18 alcohols, 8 acids,

3 esters, 14 terpenes, and 14 alkanes (Table 3). According to Ordóez et al. [38], the majority of these compounds originate from lipid auto-oxidation and bacterial metabolism (esterification by *Staphylococcus* spp., lipid oxidation, carbohydrate fermentation, and amino acid catabolism).

**Table 3.** Volatile compounds identified in beef sausage subjected to NaCl partial replacement with KCl coupled with high-pressure processing (HPP) under cold storage.

| Compounds | | 2-Octanol Equivalent Concentration (ng/g) | | | | | | |
|---|---|---|---|---|---|---|---|---|
| Aldehyde | LRI (DB624) | 100 No HPP | 100 HPP | 75:25 No HPP | 75:25 HPP | 50:50 No HPP | 50:50 HPP | Odor Class |
| Benzaldehyde | 1105 | 12.37 ± 0.02 [e] | 35.21 ± 0.06 [c] | 30.32 ± 0.00 [d] | 40.04 ± 0.18 [b] | 31.62 ± 0.03 [d] | 52.52 ± 0.08 [a] | cherry, alcohol |
| Phenylace taldehyde | 1161 | 9.49 ± 0.60 [e] | 11.51 ± 0.03 [d] | 15.88 ± 0.00 [c] | 16.56 ± 0.14 [c] | 18.10 ± 0.07 [b] | 20.26 ± 0.13 [a] | flower, malt |
| Nonanal | 1168 | 4.26 ± 0.03 [e] | 7.69 ± 0.07 [c] | 5.32 ± 0.11 [d] | 8.71 ± 0.11 [bc] | 9.10 ± 0.02 [b] | 12.80 ± 0.05 [a] | alkane |
| Pentadecanal | 1616 | 30.29 ± 0.08 [e] | 50.88 ± 0.07 [d] | 50.94 ± 0.02 [d] | 90.34 ± 0.15 [c] | 130.0 ± 0.11 [b] | 141.13 ± 0.35 [a] | flower, fresh |
| Trans-2-octenal | 1156 | 0.11 ± 0.00 [f] | 0.23 ± 0.02 [e] | 1.56 ± 0.03 [d] | 5.45 ± 0.05 [b] | 3.99 ± 0.11 [c] | 10.46 ± 0.07 [a] | fat |
| Heptanal | 1044 | 0.07 ± 0.00 [e] | 0.18 ± 0.05 [d] | 1.66 ± 0.07 [c] | 2.81 ± 0.02 [b] | 2.67 ± 0.11 [b] | 5.84 ± 0.19 [a] | citrus, fat |
| Hexadecanal | 1520 | 4.68 ± 0.18 [d] | 12.86 ± 0.37 [c] | 6.70 ± 0.02 [d] | 21.23 ± 0.22 [b] | 23.30 ± 0.19 [b] | 31.40 ± 0.35 [a] | fat, grass |
| Decanal | 1221 | 1.25 ± 0.02 [d] | 1.52 ± 0.05 [d] | 1.38 ± 0.03 [d] | 4.25 ± 0.07 [b] | 2.46 ± 0.03 [c] | 6.55 ± 0.18 [a] | green |
| Octanal | 1111 | 1.10 ± 0.00 [d] | 1.53 ± 0.02 [cd] | 1.99 ± 0.05 [c] | 2.94 ± 0.15 [b] | 1.76 ± 0.08 [c] | 4.48 ± 0.14 [a] | lemon |
| Octadecanal | 1613 | 0.96 ± 0.02 [f] | 1.69 ± 0.13 [e] | 2.39 ± 0.11 [d] | 7.96 ± 0.35 [b] | 5.61 ± 0.08 [c] | 11.94 ± 0.07 [a] | oil |
| 2-Decenal | 1264 | 0.20 ± 0.05 [d] | 0.47 ± 0.02 [e] | 0.26 ± 0.03 [d] | 0.84 ± 0.08 [b] | 0.57 ± 0.15 [c] | 1.12 ± 0.05 [a] | fat |
| 3,5-ditert-butyl-4-hydroxyben zaldehyde | 1545 | 0.21 ± 0.03 [d] | 0.49 ± 0.07 [c] | 0.27 ± 0.05 [d] | 0.95 ± 0.08 [b] | 0.73 ± 0.11 [bc] | 1.47 ± 0.03 [a] | spice, flower |
| **Total Aldehydes** | | **64.99 ± 1.78 [f]** | **124.26 ± 1.98 [d]** | **118.67 ± 3.56 [e]** | **202.08 ± 2.13 [c]** | **229.91 ± 2.38 [b]** | **299.97 ± 1.82 [a]** | |
| **Phenols** | | | | | | | | |
| Myristicin | 1401 | 186.11 ± 0.06 [a] | 2.20 ± 0.36 [d] | 1.58 ± 0.15 [d] | 8.04 ± 0.21 [c] | 7.91 ± 0.08 [c] | 12.22 ± 0.39 [b] | warm |
| Elemicin | 1423 | 84.09 ± 0.96 [a] | 24.35 ± 0.05 [d] | 25.32 ± 0.77 [d] | 32.54 ± 0.64 [c] | 52.47 ± 1.45 [b] | 54.65 ± 0.94 [b] | flower |
| Isoelemicin | 1474 | ND | 1.59 ± 0.13 [c] | 1.00 ± 0.02 [d] | 2.69 ± 1.06 [b] | 3.12 ± 0.54 [ab] | 3.95,± 0.71 [a] | flower |
| Safrole | 1268 | 143.86 ± 1.35 [a] | 10.01 ± 0.08 [c] | 10.92 ± 0.15 [c] | ND | 3.23 ± 0.07 [d] | 82.21 ± 0.13 [b] | spice, sweet |
| **Total phenols** | | **414.06 ± 1.88 [a]** | **38.15 ± 0.07 [e]** | **38.82 ± 1.78 [e]** | **43.27 ± 0.03 [d]** | **66.73 ± 0.15 [c]** | **153.03 ± 2.12 [b]** | |
| **Esters** | | | | | | | | |
| Ethyl hexanoate | 1095 | 0.12 ± 0.00 [d] | 0.35 ± 0.03 [c] | 0.41 ± 0.07 [bc] | 0.57 ± 0.11 [b] | 2.58 ± 0.08 [a] | 3.10 ± 0.07 [a] | fruit |
| Ethyl octanoate | 1203 | 2.22 ± 0.14 [c] | 2.35 ± 0.07 [c] | 3.12 ± 0.15 [b] | 3.40 ± 0.14 [b] | 3.42 ± 0.02 [b] | 4.60 ± 0.19 [a] | fruit |
| Linalyl acetate | 1169 | 1.35 ± 0.05 [c] | 1.71 ± 0.13 [c] | 2.10 ± 0.08 [b] | 2.57 ± 0.02 [b] | 4.2 ± 0.22 [a] | 4.70 ± 0.26 [a] | fruit |
| **Total esters** | | **3.69 ± 0.22 [f]** | **4.41 ± 0.19 [e]** | **5.63 ± 0.38 [d]** | **6.54 ± 0.59 [c]** | **10.2 ± 0.70 [b]** | **12.4 ± 0.14 [a]** | |
| **Alcohols** | | | | | | | | |
| Sabinene hydrate | 1173 | ND | 0.39 ± 0.11 [e] | 0.66 ± 0.02 [d] | 0.97 ± 0.00 [b] | 0.77 ± 0.07 [cd] | 1.18 ± 0.14 [a] | turpentine |
| α-Terpineol | 1223 | ND | 2.40 ± 0.05 [ab] | 2.14 ± 0.11 [ab] | 2.27 ± 0.15 [a] | 2.04 ± 0.02 [b] | 2.28 ± 0.13 [a] | mint |
| Dextro-2,3-butanediol | 1040 | ND | 0.35 ± 0.02 [a] | ND | ND | ND | ND | alcohol |
| Terbutol | 1372 | 50.41 ± 1.28 [e] | 216.37 ± 0.92 [a] | 80.92 ± 0.47 [d] | 198.29 ± 0.15 [ab] | 100.08 ± 0.78 [c] | 219.20 ± 0.36 [a] | herb |
| Anethol | 1271 | 104.12 ± 1.14 [a] | 1.88 ± 0.48 [e] | 2.16 ± 0.02 [e] | 14.44 ± 0.39 [c] | 12.98 ± 0.26 [d] | 20.09 ± 0.38 [b] | alcohol |
| Methyl eugenol | 1340 | 104.88 ± 0.82 [a] | 27.11 ± 0.71 [d] | 27.42 ± 1.08 [d] | 42.11 ± 0.14 [c] | 45.57 ± 0.58 [c] | 75.58 ± 0.41 [b] | sweet, flower |
| Methyl isoeugenol | 1393 | 92.84 ± 0.07 [a] | ND | ND | ND | ND | ND | sweet, flower |
| Eugenol | 1327 | 0.08 ± 0.02 [d] | 0.89 ± 0.05 [a] | 0.92 ± 0.11 [a] | 0.11 ± 0.07 [d] | 0.23 ± 0.11 [c] | 0.58 ± 0.11 [b] | honey, flower |
| Isomethyleugenol | 1367 | 0.20 ± 0.08 [d] | 0.35 ± 0.15 [cd] | 0.59 ± 0.08 [c] | 19.35 ± 0.21 [b] | 19.74 ± 0.07 [b] | 28.92 ± 0.97 [a] | flower, wine |
| p-Cresol | 1225 | 0.29 ± 0.02 [d] | 1.14 ± 0.11 [b] | 0.43 ± 0.00 [cd] | 1.23 ± 0.08 [b] | 0.69 ± 0.05 [c] | 2.48 ± 0.14 [a] | smoke |
| cis-β-terpineol | 1151 | 4.21 ± 0.05 [e] | 8.91 ± 0.13 [c] | 6.53 ± 0.05 [d] | 10.34 ± 0.15 [b] | 8.64 ± 0.08 [c] | 14.92 ± 0.21 [a] | alcohol |
| 1-Hexanol | 1041 | 1.24 ± 0.11 [e] | 3.96 ± 0.02 [c] | 1.57 ± 0.18 [e] | 5.64 ± 0.25 [b] | 2.71 ± 0.08 [d] | 10.71 ± 0.29 [a] | alcohol, green |
| Linalool | 1169 | 1.33 ± 0.03 [d] | 1.49 ± 0.14 [cd] | 1.81 ± 0.05 [c] | 1.58 ± 0.13 [c] | 2.01 ± 0.00 [b] | 3.79 ± 0.08 [a] | lavender, flower |
| (−)-4-terpineol | 1205 | 4.97 ± 0.02 [c] | 6.39 ± 0.07 [bc] | 6.20 ± 0.11 [bc] | 6.94 ± 0.13 [b] | 6.10 ± 0.02 [bc] | 8.69 ± 0.14 [a] | nutmeg |
| p-Cymen-8-ol | 1234 | ND | ND | 0.80 ± 0.13 [b] | 0.97 ± 0.03 [a] | ND | ND | citrus |
| (E)-Ocimenol | 1223 | 1.12 ± 0.08 [d] | 1.48 ± 0.14 [cd] | 1.97 ± 0.07 [c] | 2.11 ± 0.15 [bc] | 2.50 ± 0.14 [b] | 3.06 ± 0.02 [a] | fresh |
| Terpinen-4-ol | 1205 | 3.11 ± 0.05 [d] | 6.01 ± 0.13 [c] | 6.52 ± 0.11 [c] | 8.43 ± 0.19 [b] | 8.44 ± 0.29 [b] | 10.51 ± 0.33 [a] | nutmeg |

**Table 3.** *Cont.*

| | LRI | | | | | | | |
|---|---|---|---|---|---|---|---|---|
| **Total alcohols** | | **370.1 ± 0.78** [b] | **285.27 ± 2.17** [d] | **138.67 ± 4.08** [f] | **320.68 ± 5.56** [c] | **218.8 ± 4.56** [e] | **409.5 ± 1.01** [a] | |
| **Acids** | | | | | | | | |
| Octanoic acid | 1248 | 1.30 ± 0.07 [c] | 9.29 ± 0.15 [b] | 0.83 ± 0.03 [d] | 9.14 ± 0.08 [b] | 9.11 ± 0.21 [b] | 10.97 ± 0.18 [a] | cheese, sweat, sour |
| Acetic acid | 802 | 10.72 ± 0.36 [e] | 39.64 ± 0.52 [d] | 0.55 ± 0.02 [f] | 42.49 ± 0.25 [c] | 49.90 ± 0.45 [b] | 52.39 ± 0.47 [a] | |
| Heptanoic acid | 1198 | 0.16 ± 0.00 [f] | 0.48 ± 0.02 [c] | 0.24 ± 0.03 [e] | 0.64 ± 0.07 [b] | 0.38 ± 0.03 [d] | 0.95 ± 0.02 [a] | fat, citrus |
| Myristic acid | 1531 | ND | 2.06 ± 0.05 [c] | 0.92 ± 0.05 [d] | 3.66 ± 0.14 [b] | 0.17 ± 0.13 [e] | 4.61 ± 0.08 [a] | cheese |
| Nonanoic acid | 1296 | ND | 3.93 ± 0.08 [e] | 8.09 ± 0.05 [c] | 5.24 ± 0.13 [d] | 11.13 ± 0.02 [b] | 12.26 ± 0.38 [a] | fat |
| Decanoic acid | 1345 | 5.20 ± 0.05 [c] | 8.72 ± 0.14 [b] | 8.61 ± 0.11 [b] | 8.99 ± 0.07 [ab] | 8.52 ± 0.05 [bc] | 9.64 ± 0.15 [a] | fat |
| Pentanoic acid | 1149 | 1.32 ± 0.11 [f] | 3.15 ± 0.02 [c] | 1.54 ± 0.11 [e] | 4.61 ± 0.03 [b] | 2.34 ± 0.05 [d] | 6.66 ± 0.14 [a] | cheese |
| Hexanoic acid | 1150 | 12.11 ± 0.26 [e] | 19.04 ± 0.24 [d] | 173.85 ± 0.35 [a] | 164.07 ± 0.53 [b] | 154.84 ± 0.08 [c] | 163.47 ± 1.23 [b] | green |
| **Total acids** | | **29.51 ± 0.11** [e] | **77.02 ± 0.52** [d] | **193.8 ± 0.91** [c] | **229.7 ± 0.74** [b] | **227.28 ± 1.05** [b] | **249.98 ± 1.75** [a] | |
| **Terpenes** | | | | | | | | |
| γ-Terpinene | 1108 | ND | 0.88 ± 0.02 [c] | 1.06 ± 0.07 [b] | 0.77 ± 0.10 [d] | 0.82 ± 0.02 [c] | 1.50 ± 0.13 [a] | lemon |
| d-Limonene | 1087 | ND | 2.75 ± 0.05 [c] | 3.15 ± 0.19 [b] | 3.60 ± 0.08 [a] | 3.56 ± 0.13 [a] | 3.54 ± 0.02 [a] | fruit, mint |
| m-Cymene | 1099 | 1.07 ± 0.07 [d] | 2.01 ± 0.05 [cd] | 1.17 ± 0.00 [d] | 2.22 ± 0.02 [c] | 2.85 ± 0.03 [b] | 3.05 ± 0.08 [a] | gasoline, citrus |
| Styrene | 1024 | 5.65 ± 0.03 [e] | 12.61 ± 0.19 [d] | 16.27 ± 0.37 [c] | 17.13 ± 0.71 [b] | 16.12 ± 0.02 [c] | 18.24 ± 0.52 [a] | gasoline, citrus |
| o-Cymene | 1100 | 5.15 ± 0.02 [d] | 5.67 ± 0.07 [c] | 6.25 ± 0.19 [b] | 6.52 ± 0.35 [b] | 7.31 ± 052 [a] | 7.61 ± 0.35 [a] | fruit, gasoline |
| 4-Carene | 1108 | 14.86 ± 0.18 [c] | 26.22 ± 0.21 [a] | 26.21 ± 0.02 [a] | 24.27 ± 0.08 [bc] | 25.92 ± 0.43 [b] | 26.34 ± 0.57 [a] | fruit, lemon |
| 2-Carene | 1125 | 1.85 ± 0.08 [e] | 6.20 ± 0.21 [d] | 6.20 ± 0.02 [d] | 8.26 ± 0.35 [b] | 7.91 ± 0.53 [c] | 9.32 ± 0.72 [a] | lemon |
| pseudo-Limonene | 1093 | 9.13 ± 0.27 [c] | 52.20 ± 0.58 [b] | 53.20 ± 0.53 [b] | 55.92 ± 0.21 [b] | 61.04 ± 0.91 [a] | 59.62 ± 1.26 [ab] | fruit, citrus, mint |
| α-Thujene | 1073 | 11.95 ± 0.16 [d] | 20.52 ± 0.39 [c] | 29.07 ± 0.47 [b] | 30.08 ± 027 [ab] | 29.75 ± 0.32 [b] | 32.69 ± 0.57 [a] | green, herb |
| β-phellandrene | 1093 | 4.19 ± 0.11 [d] | 4.69 ± 0.24 [d] | 15.39 ± 0.02 [c] | 15.84 ± 0.18 [c] | 16.99 ± 0.23 [b] | 17.89 ± 0.35 [a] | spice, mint |
| p-Xylene | 997 | 45.02 ± 0.52 [d] | 62.12 ± 0.19 [c] | 83.71 ± 0.55 [b] | 85.01 ± 0.33 [ab] | 85.29 ± 0.37 [ab] | 86.46 ± 1.76 [a] | anise |
| α-Terpinene | 1125 | 0.11 ± 0.08 [e] | 0.21 ± 0.38 [d] | 0.32 ± 0.27 [c] | 0.41 ± 0.57 [b] | 0.45 ± 0.03 [b] | 0.52 ± 0.04 [a] | pine |
| β-elemene | 1289 | 3.61 ± 0.07 [d] | 4.36 ± 0.15 [c] | 5.32 ± 0.08 [bc] | 5.85 ± 0.13 [b] | 6.39 ± 0.18 [ab] | 6.83 ± 0.92 [a] | fresh, herb |
| δ-elemene | 1255 | 5.51 ± 0.03 [d] | 6.47 ± 0.21 [c] | 7.43 ± 0.02 [b] | 7.86 ± 0.32 [b] | 8.40 ± 0.32 [a] | 8.84 ± 0.51 [a] | herb |
| **Total terpenes** | | **108.1 ± 5.22** [e] | **203.28 ± 2.62** [d] | **224.33 ± 3.02** [c] | **259.37 ± 5.55** [ab] | **268.42 ± 3.81** [ab] | **277.41 ± 2.63** [a] | |
| **Ketones** | | | | | | | | |
| 2-Butanone | 388 | 7.69 ± 0.05 [a] | 6.99 ± 0.15 [b] | 6.15 ± 0.21 [bc] | 5.09 ± 0.43 [cd] | 5.51 ± 0.52 [c] | 5.04 ± 0.27 [d] | fragrant, fruit |
| 2-Nonanone | 1164 | 21.90 ± 0.31 [a] | 19.18 ± 0.21 [b] | 17.80 ± 0.13 [c] | 15.90 ± 0.27 [d] | 14.89 ± 0.28 [e] | 14.62 ± 0.33 [e] | fragrant, fruit |
| **Total ketones** | | **29.59 ± 0.36** [a] | **26.17 ± 0.58** [b] | **23.95 ± 0.43** [c] | **20.99 ± 0.58** [d] | **20.4 ± 0.52** [d] | **19.66 ± 0.41** [e] | |
| **Alkanes** | | | | | | | | |
| Heptacosane | 1693 | 1273.38 ± 52.38 [a] | 0.10 ± 0.02 [b] | ND | ND | ND | ND | alkane |
| Tetracosane | 1737 | 2604.57 ± 8.52 [b] | ND | ND | ND | ND | ND | alkane |
| Pentacosane | 1829 | 433.81 ± 1.92 [a] | 2.26 ± 0.03 [b] | ND | ND | ND | ND | alkane |
| Hexacosane | 1948 | 2483.44 ± 3.5.8 [a] | ND | ND | ND | ND | ND | alkane |
| Docosane | 1647 | 9.33 ± 0.26 [b] | 10.39 ± 0.18 [a] | ND | ND | ND | ND | alkane |
| Dodecane | 1171 | 0.21 ± 0.05 [b] | 0.71 ± 0.13 [a] | 0.40 | ND | ND | ND | alkane |
| Eicosane | 1557 | 0.57 ± 0.03 [b] | 1.46 ± 0.08 [a] | ND | ND | ND | ND | alkane |
| Heneicosane | 1602 | 2.17 ± 0.37 [b] | 3.90 ± 0.76 [a] | ND | ND | ND | ND | alkane |
| Tetracosane | 1734 | 30.85 ± 0.35 [b] | 40.31 ± 1.54 [a] | ND | ND | ND | ND | alkane |
| Triacontane | 1874 | 19.63 ± 0.72 [c] | 26.13 ± 0.33 [b] | 34.08 ± 0.15 [a] | ND | ND | ND | alkane |
| Tricosane | 1690 | 15.27 ± 0.24 [b] | 19.42 ± 0.05 [a] | ND | ND | ND | ND | alkane |
| Heptacosane | 1588 | ND | ND | ND | ND | ND | ND | alkane |
| Pentacosane | 1595 | 11.50 ± 0.32 [b] | ND | 16.39 ± 0.24 [a] | ND | 8.48 ± 0.18 [c] | ND | alkane |
| Tetradecane | 1273 | 2.34 ± 0.13 [b] | ND | ND | ND | 4.70 ± 0.03 [a] | ND | alkane |
| **Total alkanes** | | **6887.08 ± 5.48** [a] | **104.68 ± 2.31** [b] | **50.87 ± 2.15** [c] | **ND** | **13.18 ± 0.15** [d] | **ND** | |

Data were presented in mean ± SD. Different lower case letters indicate significant differences in each row at $p < 0.05$ in Duncan's multiple range tests using two-way ANOVA. ND represent not detected. Concentration expressed as ng compound in the HS/g of beef sausage. LRI: Linear retention indexes, calculated in relation to the retention time of n-alkane ($C_1$–$C_{19}$) series. Linear retention indices (LRI) of the compounds or standards eluted from the GC-MS using a DB-624 capillary column.

### 3.6.1. Effects of Partial Substitution of Sodium Chloride (NaCl) with Potassium Chloride (KCl) and HPP on Aldehyde Compounds

A high percentage of compounds were significantly ($p < 0.05$) affected by the treatments (Table 3). A total of 14 aldehyde compounds, including benzaldehyde, phenylacetaldehyde, nonanal, pentadecanal, trans-2-octenal, heptanal, hexadecanal, decanal, octanal, octadecanal, 2-decenal, and 3,5-ditert-butyl-4-hydroxybenzaldehyde, were significantly affected by both the application of HPP and the combined partial substitution of NaCl by KCl and HPP. Compared with the control (100% NaCl-no HPP), the HPP-treated samples with partial substitution of NaCl by KCl at 50%, 25%, and 0% increased the concentration of aldehyde compounds by 78.33%, 67.84%, and 47.70%, respectively. However, the untreated pressure samples treated with partial substitution of NaCl by KCl at 50% and 25% also increased the concentration of aldehyde compounds by 71.73% and 45.23% compared to the control.

### 3.6.2. Effects of Partial Substitution of NaCl with KCl and HPP on Phenol Compounds

Four phenolic compounds were significantly affected by HPP and partial substitution of NaCl by KCl independently (Table 3). Samples treated with HPP had lower contents of myristicin, elemicin, isoelemicin, and safrole. Compared to the control (100% NaCl-no HPP), the HPP treated samples with partial substitution of NaCl with KCl at 50%, 25% and 0% decreased the concentration of phenolic compounds by 63.04%, 89.55%, and 90.79%, respectively. The samples treated without pressure with partial substitution of NaCl by KCl at 50% and 25% also decreased the concentration of phenolic compounds by 83.88% and 90.62% compared to the control.

### 3.6.3. Effects of Partial Substitution of NaCl with KCl and HPP on Esters

A few percentages of the compounds were significantly ($p < 0.05$) affected by the treatments (Table 3). For all three ester compounds, including ethyl hexanoate, ethyl octanoate, and linalyl acetate, HPP and partial substitution of NaCl with KCl had an effect. Compared to the control (100% NaCl-no HPP), the HPP-treated samples with partial substitution of NaCl by KCl at 50%, 25%, and 0% increased the concentration of ester compounds by 70.24%, 43.58%, and 16.14%, respectively. However, the pressureless treated samples with partial substitution of NaCl by KCl at 50% and 25% also increased the concentration of ester compounds by 63.82% and 34.46% compared to the control.

### 3.6.4. Effects of Partial Substitution of NaCl with KCl and HPP on Alcohols

In all 18 alcohol compounds, including sabinene hydrate, α-terpineol, dextro-2,3-butanediol, terbutol, anethole, methyl eugenol, methyl isoeugenol, eugenol, isomethyleugenol, p-cresol, cis-β-terpineol, 1-hexanol, linalool, (−)-4-terpineol, p-cymen-8-ol, (E)-ocimenol, and terpinen-4-ol were detected in the beef sausages and significantly ($p < 0.05$) affected by the independent and combined treatments. Compared with the control (100% NaCl-no HPP), the concentration of alcohol compounds increased by 9.62% in the HPP-treated samples in which NaCl was partially replaced by KCl (50%). However, the HPP treated samples with partial replacement of NaCl by KCl at 25% and 0% decreased the concentration of alcohol compounds by 13.35% and 22.92%, respectively (Table 3). However, the samples treated without pressure with partial substitution of NaCl by KCl at 50% and 25% also decreased the concentration of alcohol compounds by 40.88% and 62.53% compared to the control.

### 3.6.5. Effects of Partial Substitution of NaCl with KCl and HPP on Acids

All eight acid compounds, octanoic acid, acetic acid, heptanoic acid, myristic acid, nonanoic acid, decanoic acid, pentanoic acid, and hexanoic acid were detected in the beef sausage and significantly ($p < 0.05$) affected by independent and combined treatment. Compared with the control (100% NaCl- without HPP), the HPP-treated samples with partial substitution of NaCl by KCl at 50%, 25%, and 0% increased the concentration of

acid compounds by 88.19%, 87.15%, and 61.70%, respectively (Table 3). The pressureless treated samples with partial substitution of NaCl by KCl at 50% and 25% also increased the concentration of acidic compounds by 86.82% and 84.77% compared to the control.

### 3.6.6. Effects of Partial Substitution of NaCl with KCl and HPP on Terpenes

A total of 14 terpene compounds including γ-terpinene, d-limonene, m-cymene, styrene, o-cymene, 4-carene, 2-carene, pseudo-limonene, α-thujene, β-phellandrene, p-xylene, α-terpinene, β-elemene, and δ-elemene were detected in beef sausage and significantly ($p < 0.05$) affected by the independent and combined treatments. Compared to the control (100% NaCl-no HPP), the HPP-treated samples with partial replacement of NaCl with KCl at 50%, 25% and 0% increased the concentration of terpene compounds by 61.03%, 58.32% and 46.82%, respectively (Table 3). However, the pressureless treated samples with partial substitution of NaCl by KCl at 50% and 25% also increased the concentration of terpene compounds by 59.73% and 51.81% compared to the control.

### 3.6.7. Effects of Partial Substitution of NaCl with KCl and HPP on Ketones

All two ketone compounds, 2-butanone and 2-nonanone, were detected in beef sausage and significantly ($p < 0.05$) affected by independent and combined treatments. Compared with the control (100% NaCl-no HPP), the concentration of ketone compounds decreased by 33.60%, 29.06%, and 11.56% in HPP-treated samples in which NaCl was partially replaced by KCl (50%, 25%, and 0%), respectively (Table 3). However, the pressureless treated samples with partial substitution of NaCl by KCl at 50% and 25% also decreased the concentration of ketone compounds by 31.06% and 19.06% compared to the control.

### 3.6.8. Effects of Partial Substitution of NaCl with KCl and HPP on Alkanes

A total of 14 alkanes compounds including heptacosane, tetracosane, pentacosane, hexacosane, docosane, dodecane, eicosane, heneicosane, tetracosane, triacontane, tricosane, heptacosane, pentacosane, and tetradecane were detected in beef sausage and significantly ($p < 0.05$) affected by independent and combined treatment. The combined treatments of partial replacement of NaCl by KCl and HPP strongly inhibited the alkane compounds. Compared to the control (100% NaCl-no HPP), the HPP treated samples with partial substitution of NaCl by KCl at 0% decreased the concentration of alkanes by 98.48%. The samples treated without pressure with partial substitution of NaCl by KCl at 50% and 25% also decreased the concentration of alkane compounds by 99.81% and 99.26% compared to the control (Table 3).

### 3.7. Odour Activity Value Analysis

Flavor perception is related to volatile component concentration as well as threshold levels. Figure 6 shows a heat map of 36 volatile compounds with significant differences across the six treatments. Compounds having an OAV > 1 are thought to be the main flavor producers in samples.

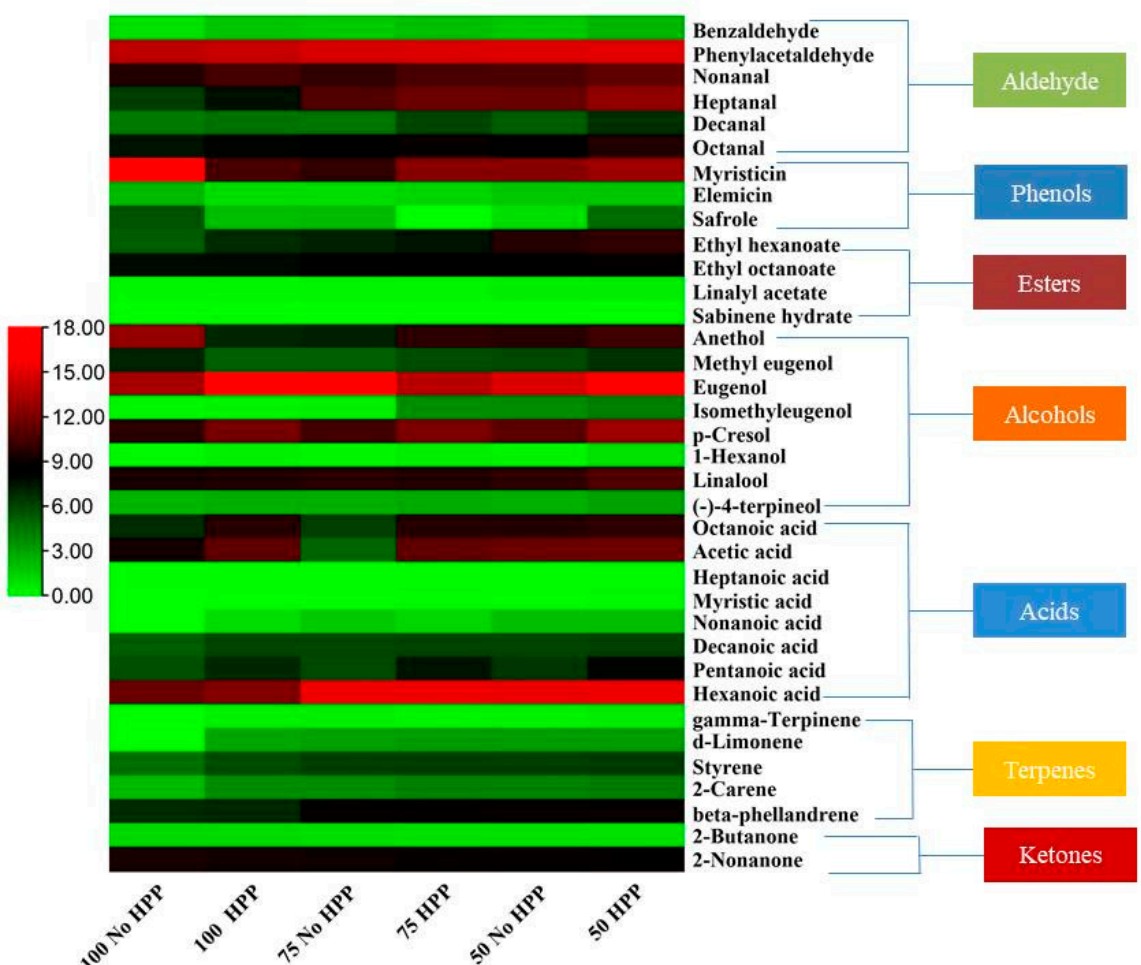

**Figure 6.** Heat map of odor activity values (OAVs) of volatile compounds in sausages refrigerated for 28 days with partial substitution of NaCl by KCl at 0%, 25%, and 50% with HPP treatment. The color scale indicates the intensity of the OAV.

## 4. Discussion

Among the substitutes for table salt (NaCl), potassium chloride (KCl) is the most widely used [39]. However, the replacement of salt with KCl in most foods must be partial or limited to 30% because higher amounts cause a bitter and metallic aftertaste [40,41]. Previous studies have shown that it is possible to replace 30% to 40% NaCl with KCl in fermented sausages [42]. In this current study, 25% and 50% of NaCl were partially replaced by KCl, combined with pressurization at 100 MPa for 5 min, which had no adverse effects on sensory properties such as taste and texture and did not significantly affect proteolysis (data not shown). Our results are in agreement with those of Wu et al. [43], who investigated the partial replacement of NaCl with KCl in bacon samples. Three salt treatments were studied: I (100% NaCl), II (60% NaCl and 40% KCl), and III (30% NaCl and 70% KCl). In general, sensory results indicate that it was possible to reduce 40% NaCl without negative effects on sensory properties such as taste and texture and without significantly affecting proteolysis. However, HPP application in our study increased the possibility of NaCl reduction to 50% with KCl without affecting the salt taste of the beef sausage. Thus, there is evidence that high pressure can cause a natural increase in salinity and thus an alternative to salt reduction [44].

Meat and meat products are considered the second-largest source of sodium in the diet, followed by baked goods. For some time, the meat industry has been using high hydrostatic pressure to improve its products [45]. Its use in meat leads to microbiological stability and changes in the functional properties of proteins, such as absorption and water binding,

improvement of emulsification ability, and solubilization of myofibrillar proteins [46]. Some studies have been conducted to investigate the interaction between high pressure and salinity on the functional properties of meat products [46]. High-pressure processing has excellent potential as a complementary technology to reduce salinity and extend the shelf life of the product. Indeed, O'Flynn et al. [47] compared the effect of pressure treatment of pork in the production of low sodium sausages. The treatment consisted of exposing the sample to different salt contents (0.5%, 1.0%, 1.5%, 2.0%, and 2.5%) at 150 MPa/5 min. Salinity levels below 1.5% had negative effects on the color, texture, juiciness, and firmness of the products. However, the results show that treatment at 150 MPa and salt contents above 2% have the potential to produce sausages without negative effects on sensory properties. The development of low-sodium meat products suggests the existence of a multifactorial process in which interactions between HPP parameters (pressure, temperature, and time), salinity, and additive concentration influence the functional properties of meat products [47].

In the present study, the substitution of 50% NaCl with KCl in combination with HPP at 100 MPa resulted in lower pH values in beef sausage stored cold for several days compared to the control sample (without pressure and 100% NaCl). The lower pH values could be attributed to the reduced NaCl content and HPP. The lower pH values of the treated HPP samples compared to the untreated samples could be beneficial to the beef sausage samples as it could inhibit pathogen growth and improve shelf-life during cold storage. The 100% and 75% NaCl with HPP treatment had higher pH values, which are not considered significant in meat product storage. Our results agree with those of [48], who found that hams subjected to HPP treatment also had higher pH than hams without HPP treatment, but the magnitude of change (<0.1 unit) may not be of practical significance.

The partial unfolding of proteins by HPP treatment after tumbling or cooking increases protein swelling and WHC [49]. The rheological properties of low-salt pork sausage were improved by pressure treatment at 200 MPa before heating [47]. The additional solubilized proteins and further gelation improved the structure and water binding. In this current study, pressurization at 100 MPa after partial replacement of NaCl with KCl at 25% and 50% had positive effects on the protein network of the beef sausages. The high NaCl concentration resulted in a loss of weight in the control samples and the sausage samples not treated with HPP, leading to lower WHC values compared to the samples with reduced NaCl combined with HPP. These results are consistent with those of Wang et al. [50], who reported that HPP at 100 MPa significantly increased WHC and formed a more regular and homogeneous three-dimensional network in rabbit myosin gel.

HPP can affect the protein structure and denaturation of the meat color pigment myoglobin and therefore has a significant effect on meat color. The extent of this effect depends on the level of pressure, the particular meat being pressurized, and the initial oxidation state of myoglobin (Mb), which are the most important factors for fresh meat. In previous studies by Horita et al. [51], no effects of 50% NaCl replacement with KCl on $L^*$, $a^*$, and $b^*$ values of mortadella with reduced fat content were found. Also in this study, 50% replacement of NaCl with KCl did not affect the $L^*$, $a^*$, and $b^*$ values of NaCl-reduced beef sausage in short days of storage at 4 °C. Partial substitution of NaCl in combination with HPP significantly affected the color of sausage during long days of storage ($p < 0.05$). Comparing samples treated with HPP from day 28 to day 0, an equal ratio of NaCl and KCl (50:50) decreased the redness ($a^*$) by 54.11%. However, the equal ratio of NaCl and KCl (50:50) coupled with HPP at day 0 increased yellowness ($b^*$) by 10.10% compared to the same ratio on day 28. Again, there was no significant difference ($p > 0.05$) among all treatments for lightness ($L^*$) with or without HPP. These results are consistent with those of Crehan, Troy, and Buckley [52], who reduced the NaCl concentration in scalded sausages from 2.5% to 1.5% and found no changes in $L^*$ values.

In this current study, storage duration, partial replacement of NaCl with KCl, and interactions with HPP treatment had significant effects on water activity (aw). Previous studies have shown that storage stability criteria with aw < 0.91 are also considered storage

stable [53]. Comparing HPP treatment at 100% NaCl with the control, the aw increased by 0.08 and 0.69 on days 3 and 21, respectively. However, on day 3, HPP-treated samples at 75% NaCl increased aw by 0.54 compared to samples without HPP. Equal ratios of NaCl and KCl (50:50) coupled with HPP treatment increased the aw value by 0.76 on day 0. An increase in KCl content increased the aw content. Therefore, the replacement of 25% and 50% NaCl with KCl could improve the water activity of beef sausages during cold storage. This result is in agreement with the findings of Hu et al. [54], who reported that for all sausages (2.5%, 2.0%, 1.5%, and 1.0%) with NaCl reduction, the aw value decreased from 0.965 to 0.734, 0.752, 0.796, and 0.824, respectively ($p < 0.05$); these differences were due to water migration inside the sausages and evaporation of surface moisture during fermentation [52]. The decrease in NaCl content significantly increased water retention from day 6 ($p < 0.05$), showing that higher NaCl content results in lower aw and moisture content [55].

Some researchers have reported that NaCl, as a pro-oxidant, may promote oxidation because it disrupts the structural integrity of the cell membrane and facilitates oxidative reactions between oxidants and unsaturated lipids in meat products. In addition, the $Na^+$-induced release of iron from heme-binding proteins may promote the rate of oxidation [6]. Several authors have observed increased lipid oxidation following HPP treatment. The descriptive effects of HPP on lipid oxidation in pork [56], beef [57], and poultry [58] have found that pressures between 300 and 600 MPa appear to be critical for triggering lipid oxidation in fresh meat. The mechanisms responsible for HPP-induced lipid oxidation are not fully understood. In general, HPP is thought to trigger lipid oxidation by two mechanisms: (1) increased accessibility of iron from hemoproteins and (2) membrane disruption. Also in this study, 100% NaCl combined with HPP treatment increased lipid oxidation in beef sausages during cold storage. This could be due to the high NaCl concentration and HPP altering the hemoprotein so that the catalytic heme group is exposed to more unsaturated fatty acids [59].

Previous studies by McArdle et al. [60] and Sun et al. [61] reported low oxidation with low TBARS values in beef and fermented mutton sausages during cold storage, which reduced the extreme quality deterioration of sausage samples throughout the storage period. Compared with our study, partial replacement of 25% and 50% NaCl with KCl in combination with HPP reduced the TBARS content in the beef sausages. This result is consistent with the findings of Wen et al. [36], who reported that the TBARS levels of one treatment (with 30% KCl) were significantly lower than those of the other treatment (with 20% KCl), which was due to the differences in KCl content.

Protein oxidation markers such as free thiols and protein carbonyls as well as free amino acids can be affected by HPP treatment at different pressures [62]. Several authors have demonstrated a link between lipid and protein oxidation in HP processed meat products [63]. In dried meat products, low water content and low water activity limit protein oxidation. In this context, Cava et al. [64] evaluated both protein (using 2,4-dinitrophenylhydrazine (DNPH) assay) and lipid oxidation (using TBARS content) in dry-cured products, pressurized at 200–300 MPa and stored at 4 °C for 90 days, and showed an increase in lipid oxidation level after HPP treatment and during storage but no significant change in protein oxidation level. These results could be partly related to the particular nature of the meat products but also the low-pressure level applied. In a recent study of vacuum-packed ground beef processed between 200 and 500 MPa, coupled monitoring of indicators of lipid and protein oxidation immediately after pressurization showed a positive correlation between the increase in hexanal content, the increase in carbonylated proteins, and the decrease in free thiols with an apparent decrease in TBARS [65]. Also in this study, the high ionic strength of NaCl and HPP increased the sulfhydryl content. However, sulfhydryl and carbonyl contents of beef sausages with partial substitution of 25% and 50% NaCl by KCl in combination with HPP decreased compared to samples without HPP treatment. The variations in sulfhydryl and carbonyl contents could be attributed to several

factors, including storage time, mild high-pressure processing technology, and substitution of less than or equal to 50% NaCl by KCl.

High salt concentration (typically NaCl > 1.5%) is usually used in sausage production to ensure technological, microbial, and sensory properties. The ability of HPP to induce protein solubility and gelation has been investigated to support or enhance acceptable meat binding in sausage making and reduce salt and phosphate content in various settings and studies [66,67]. Protein digestibility and nutritional value can be affected by food processing. Partial replacement of NaCl with KCl in conjunction with HPP significantly increased total digestibility. Increasing storage time significantly increased the total digestibility of all samples. Longer storage times could favor the degradation of amino acids as a result of prolonged storage at cold temperatures. From our results, it can be inferred that total digestibility is directly proportional to pepsin digestibility. These results are in agreement with those of Xue et al. [68], who reported that HPP significantly improved the protein digestibility of gelatinous meat products compared to samples without HPP.

A total of 75 volatile compounds were identified and quantified in the beef sausages at the end of the 28 days of storage, including 12 aldehydes, 4 phenols, 2 ketones, 18 alcohols, 8 acids, 3 esters, 14 terpenes, and 14 alkanes in our study. Most of these chemicals originated from lipid autooxidation and bacterial metabolism (esterification, lipid oxidation, carbohydrate fermentation, and amino acid catabolism), as indicated by Ordóñez et al. [38]. Lipid autooxidation is usually considered an important biochemical reaction highly associated with the development of flavor in fermented sausages [69]. Autooxidation of lipids produces mainly aliphatic aldehydes, including nonanal, octenal, hexanal, and acetaldehyde, which are related to the autooxidation of unsaturated fatty acids and enzymatic oxidation, as well as some acids (such as hexanoic acid, octanoic acid, and nonanoic acid). Aldehydes usually contribute to the unique flavor of fermented meat products because they have a low odor threshold [68]. Among them, hexanal, which contributes to fresh grass smell, was a typical product of linoleic acid oxidation [70]. However, acids do not contribute significantly to aroma due to their high odor threshold [70]. In our study, significant differences were observed in the volatile compounds of beef sausages with three different NaCl contents by partial substitution with KCl at 28 days of storage, which were well-visualized by heat map analysis. In our study, 13 compounds exhibited low OAVs (OAV < 1); however, high OAVs (OAV > 1) were obtained after partial substitution of NaCl by KCl at 25% and 50% with HPP treatment compared to the non-HPP treated samples. These compounds, including phenylacetaldehyde, nonanal, heptanal, octanal, myristicin, ethyl hexanoate, anethole, eugenol, p-cresol, linalool, octanoic acid, acetic acid, pentanoic acid, hexanoic acid, beta-phellandrene, 2-nonanone, ethyl octanoate, Safrole, methyl eugenol, isomethyl eugenol, d-limonene, and styrene (with OVA > 5), have a high impact on the overall aroma profile of beef sausages cold-stored for 28 days. Of these compounds, most are formed by lipid auto-oxidation, lipid-β-oxidation, and esterification.

## 5. Conclusions

The World Health Organization (WHO) has agreed to reduce the global population's intake of salt by 30% by 2025 [71]. Therefore, following scientific information and health recommendations, meat companies are trying to develop low-salt products to reduce dietary sodium consumption. Decreasing and replacing NaCl intake has been identified as one of the most cost-effective measures to improve the population's health, especially with KCl. In this current study, the partial substitution of NaCl with KCl in combination with HPP significantly affected pH, aw value, WHC, TBARS, protein oxidation, in vitro digestibility, and volatile compound composition of beef sausages during 28 days of storage at a temperature of 4 °C. The identified volatile compounds decreased on day 28 with increasing duration of storage. This could be due to a significant increase in lipid oxidation during storage. Therefore, partial replacement of NaCl with KCl in conjunction with HPP reduced the degree of lipid oxidation during cold storage, which had a positive effect on preventing the formation of undesirable flavor in the beef sausages. In conclusion, HPP

can be considered in NaCl-reduced or partial substitution with KCl in meat products to improve the physiochemical and flavor properties during cold storage at a temperature of 4 °C. In the future development, HPP and KCl can replace the second industrial treatment of NaCl reduction in beef sausages while maintaining the physical attributes, ensuring meat functionality and quality.

**Author Contributions:** Conceptualization, L.Z. and T.O.; methodology, T.O.; software, S.B.; validation, L.Z., F.K.A. and T.O.; formal analysis, Z.W.; investigation, Y.G.; resources, L.Z.; data curation, T.O.; writing—original draft preparation, T.O.; writing—review and editing, T.O. and S.B.; visualization, F.K.A.; supervision, L.Z.; project administration, L.Z.; funding acquisition, L.Z. All authors have read and agreed to the published version of the manuscript.

**Funding:** The study was funded by Education Science and Technology Innovation Project of Gansu Province (GSSYLXM-02).

**Institutional Review Board Statement:** Not applicable.

**Informed Consent Statement:** Not applicable.

**Data Availability Statement:** The data presented in this study are available on request from the corresponding author.

**Acknowledgments:** The authors would like to thank the laboratory team that analyzed the samples and all the students who volunteered to participate in the experiment.

**Conflicts of Interest:** The authors declare that they have no competing interest.

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
