# Peer review of "Effect of Partial Substitution of Sodium Chloride (NaCl) with Potassium Chloride (KCl) Coupled with High-Pressure Processing (HPP) on Physicochemical Properties and Volatile Compounds of Beef Sausage under Cold Storage at 4 °C"

_processes, doi:10.3390/pr10020431_

Round 1

Reviewer 1 Report

I have no objections. 

Reviewer 2 Report

The authors have to be more precise about the number of samples in each group of sausages. This is not clear from the description of the experimental design and the subsection about the sausage preparation.

The results should be better presented as mean and Standard deviation.

Reviewer 3 Report

The article aims to evaluate the effect of NaCl reduction and KCl substitution together with high pressure  in a refrigerated beef product on Aw, pH, WHC, colour, volatiles profile, and digestibility.
The paper shows a large amount of analysis performed, however the experimental design is complex and not well detailed.
The authors refer to high hydrostatic pressure with treatments of 100 MPa 5 min, what is the objective pursued with HP treatment?

The treatment conditions are far below those used to ensure the food safety of a meat food similar to the one studied, if that is the objective of the  HP treatment. If the use of HP pursues another objective, the authors should make reference to this aspect, both in the introduction and in the objective of the work.
There are a significant number of inaccuracies, errors or lack of specificity in the materials and methods section.
An experimental design is used with 14 formulations (Table 1) that are subsequently reduced to 3 formulations (100% NaCl, 75% NaCl 25% KCl and 50% NaCl 50% KCl; with/without HP) and on this number of experimental groups the work is carried out.
The application of isostatic high pressure (100 MPa 5 min) is applied at two times (line 123-126 and 140-143) while in figure 1 only one high pressure treatment is shown. 
Certain analyses are not well described (including important errors in the description). Therefore, the article requires extensive revision to correct the. 
i.e. 
lines 198-199 [...] the resulting solution was measured using a spectrophotometer (00080S, Full wave-length microplate reader, China) against a RO [...].
line 215 [...] (Sigma laboratory centrifuges 3k15, Bie and Berntsen A/S, Denmark), after which [...] [...]

There are certain sentences which do not have sense:
line 235 [...] Myofibrillar proteins? were suspended in 33 mM glycine buffer with a pH of 1.8 (gastric pH), and the final concentration was adjusted to 0.8 mg/ml.[...]
line 263 -265: [...] The temperature programme and detection method [37]. Compound were identified by injecting commercial standards, comparing spectra with the 264 Wiley7Nist05 Library [...]
The graphs need to be improved.
There is a lack of data concerning the analysis of volatiles, which internal standard is used, at what dosage and what are the extraction conditions by SPME, GC and MS.

Author Response

This manuscript is a resubmission of an earlier submission. The following is a list of the peer review reports and author responses from that submission.

Round 1

Reviewer 1 Report

My detail comments are shown in the attached file. Since the statistical analysis was incorrectly done, a major revision of the manuscript is needed.

Reviewer 2 Report

After reviewing the manuscript "Effect of partial substitution of sodium chloride (NaCl) with potassium chloride (KCl) coupled with high-pressure processing (HPP) on physicochemical properties and volatile compounds of beef sausage under cold storage at 4 ℃", I conclude that it cannot be published in Processes. According to the description of the experiment, it was performed once and the statistical calculations were made for the successive repetitions on the same product. Therefore, it cannot be concluded that the observed effect of NaCl substitution or HPP treatment will be reproducible.